# Behavioral and Electrophysiological Effects of Ajowan (*Trachyspermum ammi* Sprague) (Apiales: Apiaceae) Essential Oil and Its Constituents on Nymphal and Adult Bean Bugs, *Riptortus clavatus* (Thunberg) (Hemiptera: Alydidae)

**DOI:** 10.3390/insects11020104

**Published:** 2020-02-04

**Authors:** Sung-Chan Lee, Seon-Mi Seo, Min-Jung Huh, Jun-Hyeong Kwon, Il Nam, Ji-Hong Park, Il-Kwon Park

**Affiliations:** 1Department of Forest Sciences, College of Agriculture and Life Sciences, Seoul National University, Seoul 08826, Korea; sungchan1225@empas.com (S.-C.L.); popcon24@naver.com (S.-M.S.); shadma@snu.ac.kr (M.-J.H.); sibollr.o.k@gmail.com (J.-H.K.); namil1913@daum.net (I.N.); ined4486@snu.ac.kr (J.-H.P.); 2Research Institute of Agriculture and Life Science, College of Agriculture and Life Sciences, Seoul National University, Seoul 151-921, Korea

**Keywords:** repellent activity, ajowan essential oil, carvacrol, thymol, bean bug, *Riptortus calvatus*, push-pull strategy

## Abstract

We investigated the repellent effect of 12 Apiaceae plant essential oils on nymphal and adult (male and female) forms of the bean bug, *Riptortus clavatus* (Thunberg) (Hemiptera: Alydidae), using a four-arm olfactometer. Among the essential oils tested, ajowan (*Trachyspermum ammi* Sprague) essential oil showed the strongest repellent activity against the nymphal and adult bean bugs. For female adults, the repellent activity was significantly different between an ajowan oil-treated chamber and an untreated chamber down to a concentration of 14.15 μg/cm^2^. We also investigated the repellent activity of individual ajowan essential oil constituents. Of the compounds examined, carvacrol and thymol showed the most potent repellent activity against the nymphal and adult bean bugs. Carvacrol and thymol exhibited 73.08% and 70.0% repellent activity for the bean bug nymph at 0.71 and 2.83 μg/cm^2^, respectively, and 82.6% and 80.7% at 5.66 and 11.32 μg/cm^2^, respectively, for male adults. Carvacrol and thymol exhibited strong repellent activity against female adult bean bugs down to a concentration of 2.83 μg/cm^2^. Ajowan essential oil, thymol and carvacrol elicited a negative electroantennogram (EAG) response from adult bean bugs. This could explain the repellent activity of ajowan essential oil and its constituents. Our results indicate that ajowan essential oil and its constituents carvacrol and thymol can be potential candidates as the ‘push’ component in a ‘push-pull’ strategy for bean bug control.

## 1. Introduction

The bean bug *Riptortus clavatus* Thunberg (Heteroptera: Alydidae) has caused serious damage to soybean and tree fruits in Korea and Japan [1,2,3]. Several control methods have been developed to manage bean bugs [4]. In Japan and Korea, synthetic insecticides are usually sprayed two or three times per season to control the bean bugs [4,5]. However, spraying insecticides does not effectively control bean bugs because of their high flight activity [6]; they often reinvade the fields soon after insecticide application [7]. Furthermore, Kikuchi et al. [8] reported that *R. clavatus* showed resistance to insecticides by harboring insecticide-degrading gut symbiotic bacteria of the genus *Burkholderia*. 

Semiochemicals have been studied to better understand how to monitor and manage bean bugs [9,10,11,12]. Four compounds, (*E*)-2-hexenyl (*Z*)-3-hexenoate (*E*2H*Z*3H), (*E*)-2-hexenyl (*E*)-2-hexenoate (*E*2H*E*2H), tetradecyl (=myristyl) isobutyrate (C14:iBu) and octadecyl isobutyrate, were identified as bean bug aggregation pheromones [13,14,15]. The most effective ratio of *E*2H*Z*3H, *E*2H*E*2H, and C14:iBu was confirmed as 1:1:1 [11,12]. Although use of an aggregation pheromone trap has been considered effective, the control effect has not yet been clarified. Rahman et al. [16] reported that aggregation pheromone traps did not reduce the bean bug population or crop damage. They found that aggregation pheromone traps were not effective for controlling bean bugs and new bean bug control strategies are necessary. Providing insects with a choice between more attractive odors such as host plant volatiles or pheromones (pull) and host plants containing repellents (push) could be a good strategy for enhancing the effectiveness of current control methods. Natural repellents based on plant essential oils have already been commercialized and used for this ‘push and pull’ strategy in insect management [17,18], though there have been no reports on the repellent effect of plant essential oils and their constituents on *R. clavatus*. 

In this study, we evaluated the repellent activity of 12 Apiaceae plant essential oils and ajowan essential oil constituents against nymph and adult male and female bean bugs to select a specific repellent suitable for use in a push-pull system. 

## 2. Materials and Methods 

### 2.1. Plant Essential Oils and Chemicals

Plant essential oils from *Ammi visnaga* L., celery (*Apium graveolens* L.), coriander herb (*Coriandrum sativum* L.), coriander seed (*Coriandrum sativum* L.), carrot seed (*Daucus carota* L.), galbanum (*Ferula galbaniflua* Boiss and Bulise), pastinak (*Pastinaca sativa* L.) and parsley (*Petroselinum sativum* Hoffm) were purchased from the online retailer Oshadhi Ltd (www.oshadhi.eu). Ajowan (*Trachyspermum ammi* Sprague), dill (*Anethum graveolens* L.), caraway (*Carum carvi* L.) and cumin (*Cuminum cyminum* L.) were purchased from the online retailer Jinarome (www.jinarome.com). We obtained test compounds (+)-*α*-pinene (>99%), *β*–myrcene (95%), (+)-limonene (97%) and terpinen-4-ol (97%) from Sigma-Aldrich (Milwaukee, WI, USA). Tokyo Kasei (Tokyo, Japan) supplied camphene (80%), (+)-*β*-pinene (94%), carvacrol (95%) and *N*,*N*-diethyl-*meta*-toluamide (DEET) (98%). We purchased *α*-terpinene (85%), *p*-cymene (95%), 1,8-cineole (99%), *γ*-terpinene (97%) and thymol (99%) from Fluka (Buchs, Switzerland). (*E*)-2-hexenyl (*Z*)-3-hexenoate (purity 95.96%), a bean bug aggregation pheromone, was provided by the Korea Institute of Insect Pheromone (Daejeon, Republic of Korea). 

### 2.2. Insects

The bean bugs (*Riptortus clavatus*) were collected using aggregation pheromone traps at Seoul National University campus, Seoul, Republic of Korea in 2016 and 2017. We reared the bean bugs according to the method of Choi et al. [19] with slight modification. In brief, the bugs were maintained in an insect rearing room in a plastic cage (16.5 cm in diameter by 8 cm in height) at 25 ± 1 °C and 60% relative humidity under a 16:8 h (L:D) cycle. We supplied dried soybean seed, peanut seed and water (with 0.005% ascorbic acid) for food. We used fourth instar nymphs and 10–20-day-old adults for our bioassays. 

### 2.3. Gas Chromatography (GC)

Chemical components of ajowan essential oil were analyzed using a gas chromatograph (Agilent 7890N, Santa Clara, CA, USA) installed with a flame ionization detector (FID). We compared the retention time of the essential oil components with those of authentic compounds using DB-1MS and HP-INNOWAX columns (length: 30 m, inner diameter: 0.25 mm, film thickness: 0.25 μm, J&W Scientific Inc., Fosom, CA, USA). The oven temperature was maintained at 40 °C for 1 min, raised to 250 °C at the rate of 6 °C/min, and then held at 250 °C for 4 min. The flow rate of nitrogen (carrier gas) was 1.5 mL/min. The retention indices of ajowan essential oil constituents were determined by calculating the retention compared with those of C_8_–C_22_ n-alkanes using the same oven temperature conditions described above. 

### 2.4. Gas Chromatography-Mass Spectrometry

Gas chromatography-mass spectrometry (GC-MS) (Agilent 7890B GC, Agilent 5977B MSD, Santa Clara, CA, USA) analysis was conducted using a DB-5MS column (length: 30 m, inner diameter: 0.25 mm, film thickness: 0.25 μm, J&W Scientific Inc., Fosom, CA, USA). We used the same oven temperature conditions for gas chromatography as described above. The flow rate of helium was 1.0 mL/min. The effluent from the gas chromatography column was transported directly into the mass spectrometry source through a delivery line heated to 250 °C. The ionization of each component was accomplished using electron impact with 70 eV. We identified ajowan essential oil constituents by comparing retention indices and the mass spectra pattern of each component with those of standard samples supplied by the National Institute of Standards and Technology Mass Spectral (NIST MS) library. 

### 2.5. Repellent Activity Test

We evaluated the repellent activities of the plant essential oils and their constituents for nymphal and adult bean bugs using a four-arm olfactometer with a slight modification of a prior study [20]. The olfactometer consisted of a central chamber (7.5 cm in diameter by 5 cm in height) with four arms (2 cm in diameter by 12 cm in length) connected to glass side chambers (7.5 cm in diameter by 5 cm in height) (Figure 1A). Plant essential oils and constituents dissolved in hexane were applied to a paper disc (8 mm, Advantec MFS, Inc., Dublin, CA, USA). The paper disc control had hexane only. The hexane was allowed to evaporate for approximately 30 s in the fume hood before the paper discs were transferred to the side glass chamber. The experimental paper disc, dried soybean, and water-soaked cotton wool were placed in one set of two opposite chambers. The two opposite chambers held the control paper disc, dried soybean, and water-soaked cotton wool (Figure 1A,B). A single nymphal or adult bean bug was introduced to the central chamber and allowed to move freely. After 24 h, we counted the number of nymphs and adult bugs that stayed in the four side chambers. Any bugs that stayed in the central chamber or one of the four connecting arms were considered as a non-response. Thirty nymphs and adults were used for each bioassay. Individual nymphs and adults were used only once. To minimize the effect of direction and position, the treated and untreated paper discs were placed in opposite chambers for each bioassay. Upon each bioassay completion, the glass assay devices were cleaned with detergent and water and dried for 1 h at 100 °C in a drying oven. Test bean bugs were maintained at 25 ± 1 °C and 60% relative humidity under a 16:8 h (L:D) cycle. The repellent activity was calculated according to the following formula: repellent activity (%) = U/(T + U) × 100 (U = number of bean bugs in untreated chambers; T = number of bean bugs in treated chambers). The data were analyzed using the chi-square test (*χ*^2^ test) (IBM SPSS Statistics V25.0, Armonk, NY, USA).

### 2.6. Electroantennogram (EAG)

We used an electroantennogram (EAG) set composed of a signal acquisition meter (IDAC-4, Syntech, Buchnbach, Germany), probes (universal single ended probe, Syntech, Germany) and a stimulus controller (CS 55, Syntech, Germany) to record the electrophysiological response of the male and female adult bean bugs to ajowan essential oil, thymol, carvacrol, DEET and (*E*)-2-hexenyl (*Z*)-3-hexenoate (*E*2H*Z*3H, the bean bug’s aggregation pheromone component). Before connecting the EAG probe to the bugs, we cut the basal and tip segments of the antennas of males (n = 5) and females (n = 5). The universal probe-attached glass capillary (1.1 mm I.D., Paul Marienfeld GmbH, Germany) filled with 0.1 N KCl was connected to the basal part of the antenna and used as a recording electrode. Another probe-attached glass capillary filled with 0.1 N KCl was connected to the tip of the antenna and used as a reference electrode [21]. Test compounds (1 mg) dissolved with 10 uL of hexane were loaded onto a paper disc (8 mm, Advantec MFS, Inc.). The paper disc for control received hexane only. After the evaporation of hexane for 30 s, each test compound stimulated the antenna for 1.5 s with filtered air using a stimulus controller. Electrophysiological responses of the antenna to each test compound were acquired using a signal acquisition meter, and the magnitude of each response was measured using AutoSpike software (Syntech, Germany). The data were analyzed using ANOVA, and mean values were compared using Duncan’s test (IBM SPSS Statistics V25.0, Armonk, NY, USA). 

## 3. Results and Discussion

### 3.1. Repellent Activities of Apiaceae Plant Essential Oils

The repellent activities of 12 Apiaceae plant essential oils on nymphal and adult bean bugs are shown in Figure 2. Of the essential oils tested, ajowan, coriander herb, coriander seed, cumin, carrot seed, pastinak and parsley showed a >90% repellent activity for bean bug nymphs at 226.35 μg/cm^2^. Only the ajowan essential oil exhibited a ≥90% repellent activity at 113.18 μg/cm^2^. Bean bug nymphs showed a significant difference in repellency between the ajowan oil-treated chamber and the untreated chamber down to 28.29 μg/cm^2^, but a significant difference was not observed at 14.15 μg/cm^2^. For adult males, ajowan, *Ammi visnaga*, caraway, coriander herb, coriander seed, cumin, carrot seed and parsley essential oils showed a ≥90% repellent activity at 226.35 μg/cm^2^. Ajowan oil exhibited the strongest repellent activity down to 14.15 μg/cm^2^. No significant repellency difference was observed for other active oils at 113.18 or 56.59 μg/cm^2^. The adult female bean bugs showed a similar behavior to adult males in response to the ajowan essential oil. Strong repellency was observed down to 14.15 μg/cm^2^. Caraway, coriander herb, carrot seed and parsley essential oils exhibited a strong repellent activity for the adult female bean bugs at 113.18 μg/cm^2^, but no significant difference in repellency was observed at 56.59 μg/cm^2^. Repellent activities of plant essential oils have been mainly investigated for various biting insects and arthropods such as mosquitoes, flies, ticks and other medically important arthropod pests [22,23,24,25]. Besides their valuable use as repellents against biting insect pests, plant essential oils are an important source of stimuli for the push component of semiochemical-based ‘push-pull’ strategies [26,27]. Mauchline et al. [28] reported that lavender, *Lavendula angustifolia* Mill (Lamiaceae), essential oil caused a significant reduction in the number of pollen beetles, *Meligethes aeneus* (Fabricius) (Coleoptera: Nitidulidae), infesting oilseed rape plants in field experiments. They determined that lavender essential oil could be a valuable push component of the ‘push-pull’ system. Similarly, rosemary essential oil is a promising ‘push’ repellent against the tea green leafhopper, *Empoasca vitis* Göthe (Hemiptera: Cicadellidae) [26]. There are no reports on the repellent activity of plant essential oils on the bean bug, *R. clavatus.* In this study, we evaluated the repellent activities of 12 Apiaceae plant essential oils on nymphal and adult (male and female) bean bugs. Ajowan essential oil showed the most potent repellent activity. Many biological activities of ajowan essential oil, such as insecticidal, nematicidal and antifungal activities, have been reported [29,30,31]. Recently, Lee et al. [32] reported that ajowan essential oil exhibited potent repellent activity against adult German cockroaches. Additionally, this study and prior studies indicate that ajowan essential oil could be a potential candidate as a ‘push’ component in a ‘push-pull’ system for bean bug control. The most effective combination of aggregation pheromone components, trap type, trap color and aggregation pheromone synergists has been well studied in Korea [11,12,33,34,35,36]. 

However, the high volatility of plant essential oils must be overcome to use them as ‘push’ components in field conditions. Various controlled release formulations of bioactive compounds derived from plants have been developed in fields such as medicine, pharmaceuticals, food technology and cosmetology [37], though this is only in the beginning stages in agriculture. To actualize ajowan oil’s repellent activity in the field, the development of a suitable controlled release formulation is needed. Furthermore, the effect of ajowan essential oil on the behavior of *Gryon japonicum* and *Ooencyrtus nezarae*, effective egg parasitoids of the bean bug, should be investigated in future studies.

### 3.2. Repellent Activities of Constituents Identified from Ajowan Essential Oil

Our analysis of ajowan essential oil with GC and GC-MS agreed with a prior report [38]. The 12 most abundant compounds were identified from ajowan essential oil, and their abundance is presented in Table 1. The most abundant compound was thymol (41.77%) followed by *γ*-terpinene (27.77%), *p*-cymene (24.40%) and *β*-pinene (1.26%). The composition rate of other compounds was less than 1%. We conducted experiments on the repellent activity of individual ajowan essential oil constituents on nymphal and adult bean bugs (Figure 3). A significant difference in repellency was observed between the carvacrol-treated chamber and the untreated chamber down to 0.71 μg/cm^2^. The repellent activity of thymol was 70.0% at 2.83 μg/cm^2^, but reduced to 63.3% at 1.41 μg/cm^2^. No significant difference in repellency was observed between the chamber treated with the other ajowan essential oil constituents and the untreated chamber at 22.64 μg/cm^2^. DEET, a widely used mosquito repellent, showed an 81.4% repellent activity against bean bug nymphs at 22.64 μg/cm^2^. However, no significant difference in repellency was observed at 11.35 μg/cm^2^. Male bean bug adults exhibited repulsion to the ajowan oil constituents that was similar to what was seen in the nymphs. Strong repellent activities were observed only for the carvacrol- and thymol-treated chambers. The carvacrol repellent activity was 82.6% at 5.66 μg/cm^2^, and the rate reduced further to 59.2% at 2.83 μg/cm^2^. The male bean bug adults exhibited a strong repulsion to thymol down to 11.32 μg/cm^2^, but there was no significant change in repellency at 5.66 μg/cm^2^. Significant differences in their repulsion to DEET were seen at 22.64 μg/cm^2^, but they did not show repulsion at 11.32 μg/cm^2^. 

Female bean bug adults showed repulsion to various constituents of ajowan essential oil including *α*-pinene, 1,8-cineole, terpinen-4-ol, carvacrol and thymol at 22.64 and 11.32 μg/cm^2^. Repellent activities of *α*-pinene, 1,8-cineole, and terpinen-4-ol were 68.9%, 77.7% and 77.7%, respectively. Since the repellent activities of *α*-pinene, 1,8-cineole, and terpinen-4-ol were weaker than those of thymol and carvacrol at 22.64 μg/cm^2^, we did not investigate their repellent activity at lower concentrations. A significant difference in repellency was observed between the carvacrol- or thymol-treated chambers and the untreated chamber down to 2.83 μg/cm^2^. No significant difference in repellency was observed between the DEET-treated chamber and the untreated chamber at 22.64 μg/cm^2^. 

In our study, thymol and carvacrol showed the most potent repellent activities for nymphal and adult (male and female) bean bugs; most other ajowan essential oil constituents exhibited weak or no repellent activity. Many researchers have examined the repellent activities of thymol and carvacrol against several arthropod species. Kim et al. [39] reported that thymol and carvacrol elicited strong repellent activity for *Tribolium castaneum* (Herbst), a common stored-products insect pest, at 0.03 and 0.006 mg/cm^2^. Park et al. [40] found that thymol and carvacrol had the lowest repellent concentration (RC) values against female adult Asian tiger mosquitos (*Aedes albopictus*) of the compounds identified in *Thymus magnus*. Thymol and carvacrol repellency have also been reported for the castor bean tick, *Ixodes ricinus*, and the poultry red mite, *Dermanyssus gallinae* [41,42]. Additionally, their repellent activity is comparable to DEET, the most common active ingredient in insect repellents. For example, Tabanca et al. [43] reported that carvacrol and thymol’s repellent activity against *Ae. aegypti* was comparable to DEET. Minimum effective dosages of carvacrol, thymol and DEET were 0.013, 0.031 and 0.039 mg/cm^2^, respectively. Carvacrol exhibited similar repellent activity to DEET on *Rhodnius prolixus* and *Triatoma infestans* [44]. For this reason, this study used DEET as a positive control. DEET’s repellent activity against other hemipteran insects has also been reported in prior studies [45,46]. The thymol and carvacrol repellent activity against both the nymphal and adult bean bugs was much stronger than DEET. These results and those of prior studies indicate that thymol and carvacrol can be potential candidates as ‘push’ stimuli in push-pull strategies for bean bug control. Besides their strong repellent activity, thymol and carvacrol have several advantages as ‘push’ stimuli. First, they are listed in the ‘Generally Recognized as Safe (GRAS)’ chemical list approved by the US Food and Drug Administration (FDA) [47]. This implies that repellents based on thymol and carvacrol will not cause safety concerns when applied in the field. Second, thymol and carvacrol’s toxicity on non-target organisms such as honeybees, mealworms, beetles, shellfish and the mosquitofish, *Gambusia affinis,* was found to be negligible [48,49,50]. This lower toxicity might not disturb the foraging activity of *G. japonicum* and *O. nezarae*, the bean bug’s effective egg parasitoids. However, the effect of ajowan essential oil, thymol and carvacrol on the behavior of egg parasitoids should be investigated in future studies. Third, the long-lasting effects of thymol and carvacrol compared to other plant essential oils’ constituents are another advantage. Several studies have reported many plant-derived materials exhibiting repellent properties in recent years [51]. However, nearly all plant-based repellents derived from plant essential oils have limited residual activity (<2–4 h), primarily due to their high volatility [52,53]. Masoumi et al. [41] reported that a thymol and carvacrol mixture exhibited residual toxicity against *D. gallinae* until 14 days after spraying. Residues of thymol and carvacrol in water were 48.9% and 54.1% of their original amounts at seven days after treatment [38]. These residue rates are much higher than those of other monoterpene hydrocarbons, alcohols and aldehydes, which indicates that thymol and carvacrol are suitable candidates for a ‘push’ component. 

Our results show that ajowan plant essential oil and its components are good ‘push’ stimulus candidates for ‘push-pull’ strategies of bean bug control. Further research on developing a formulation with long-term controlled-release potential is essential for the practical use of natural repellents based on ajowan plant essential oil and its components in a ‘push and pull’ strategy. 

### 3.3. EAG Response of Bean Bugs

The mean EAG response of the male and female bean bug adults is shown in Figure 4. Mean EAG responses of the male *R. clavatus* to *E*2H*Z*3H (the bean bug’s aggregation pheromone component), ajowan essential oil, thymol, carvacrol, DEET and the blank were 0.90, 0.36, 0.29, 0.20, 0.42 and 0.25 mV, respectively. The mean EAG responses of the female bean bug to pheromone, ajowan essential oil, thymol, carvacrol, DEET and blank were 0.84, 0.61, 0.38, 0.24, 0.36 and 0.24 mV, respectively. The EAG responses to pheromone were generally higher than ajowan essential oil, thymol, carvacrol, DEET and the blank (Figure 4).

However, there was a significant difference in the shape of the EAG response waveform. The pheromone, DEET and blank elicited a positive EAG response, but ajowan essential oil, thymol and carvacrol resulted in negative EAG responses (Figure 5). 

Our study shows that there is a significant difference between compounds in the shape of the EAG responses from male and female bean bugs. Both showed positive EAG responses to the aggregation pheromone, DEET, and the blank along with negative EAG responses to the ajowan essential oil, thymol and carvacrol (Figure 5). An action potential is generated when odors depolarize or hyperpolarize the olfactory sensory neurons [54,55]. Contreras et al. [56] reported that the shape of EAG potentials correlates with the behavioral response of *Periplaneta americana*. They determined that chemicals with reversed EAG responses caused repellent activity. Ramachandran et al. [57] reported that thymol and carvacrol along with a strong anti-feeding activity elicited reversed EAG responses for two sympatric leaf folder species, *Cnaphalocrocis medinalis* and *Marasmia*. This study and prior studies indicate that ajowan essential oil, thymol, and carvacrol demonstrate repellent activity by hyperpolarizing the olfactory sensory neurons of adult bean bugs. However, the exact hyperpolarizing mechanism should be investigated further in future studies. 

## 4. Conclusions

Ajowan plant essential oils showed strong repellent activity for nymphal and adult (male and female) bean bugs, *Riptortus clavatus*. Among the constituents identified from ajowan plant essential oil, thymol and carvacrol exhibited the most potent repellent activity against bean bugs. Based on our results, we conclude that ajowan plant essential oil and its constituents carvacrol and thymol could be potential candidates as the ‘push’ component with an aggregation pheromone trap as the ‘pull’ component in a ‘push-pull’ strategy of bean bug control.

## Figures and Tables

**Figure 1 insects-11-00104-f001:**
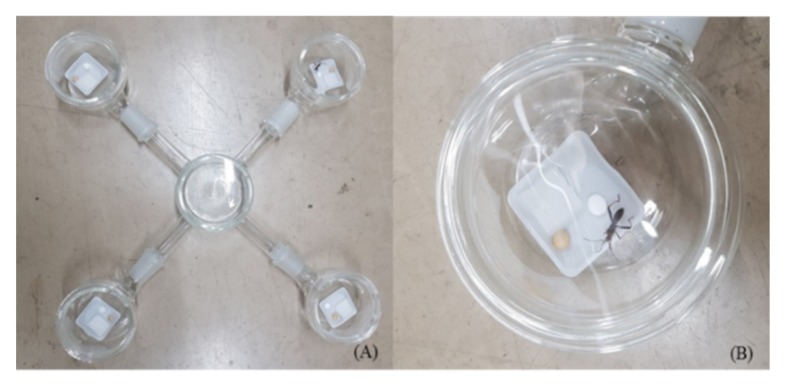
Four-arm olfactometer (**A**) and close-up image of a chamber (**B**) used to test repellent activity on the bean bug, *Riptortus clavatus*.

**Figure 2 insects-11-00104-f002:**
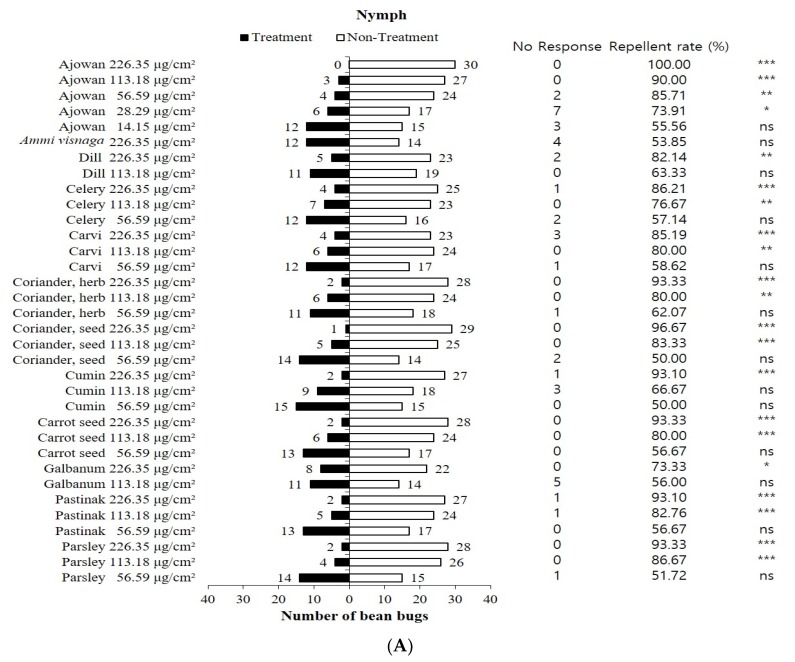
Repellent activities of 12 Apiaceae plant essential oils on nymphal (**A**) and adult (male (**B**) and female (**C**)) bean bugs. Asterisks indicate significant differences (chi-square test, *χ*^2^ test). *: *p* < 0.05, **: *p* < 0.01, ***: *p* < 0.001. ns = not significant.

**Figure 3 insects-11-00104-f003:**
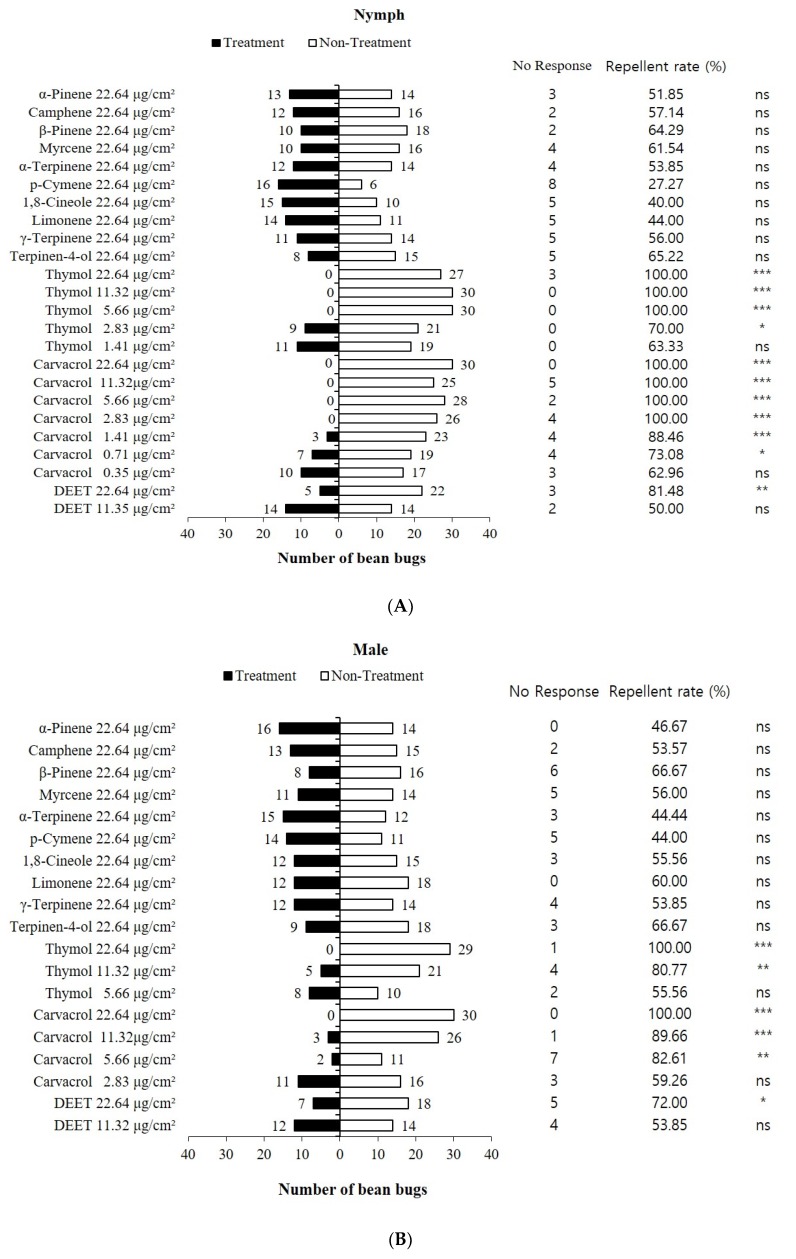
Repellent activities of constituents from ajowan essential oil on nymphal (**A**) and adult (male (**B**) and female (**C**)) bean bugs. Asterisks indicate significant differences (chi-square test, *χ*^2^ test). *: *p* < 0.05, **: *p* < 0.01, ***: *p* < 0.001. ns = not significant. DEET = *N*,*N*-Diethyl-*meta*-toluamide.

**Figure 4 insects-11-00104-f004:**
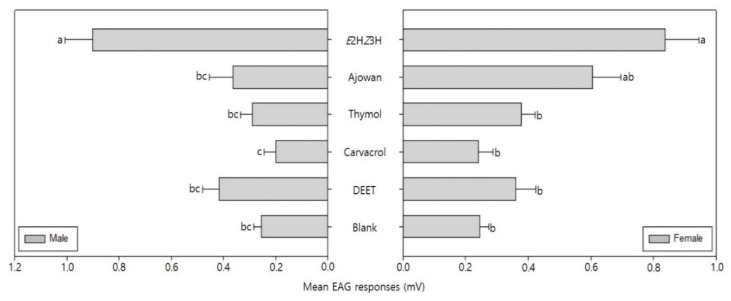
Electroantennogram (EAG) responses (mean ± S.E. (mV), n = 5) of adult bean bugs to 1 mg of (*E*)-2-hexenyl (*Z*)-3-hexenoate, ajowan essential oil, thymol, carvacrol or DEET. Different letters indicate significant differences between the mean values (F_5,24_ = 20.239, *p* < 0.001 (male, **right**), F_5,24_ = 3.98, *p* = 0.018 (female, **left**), Duncan’s test).

**Figure 5 insects-11-00104-f005:**
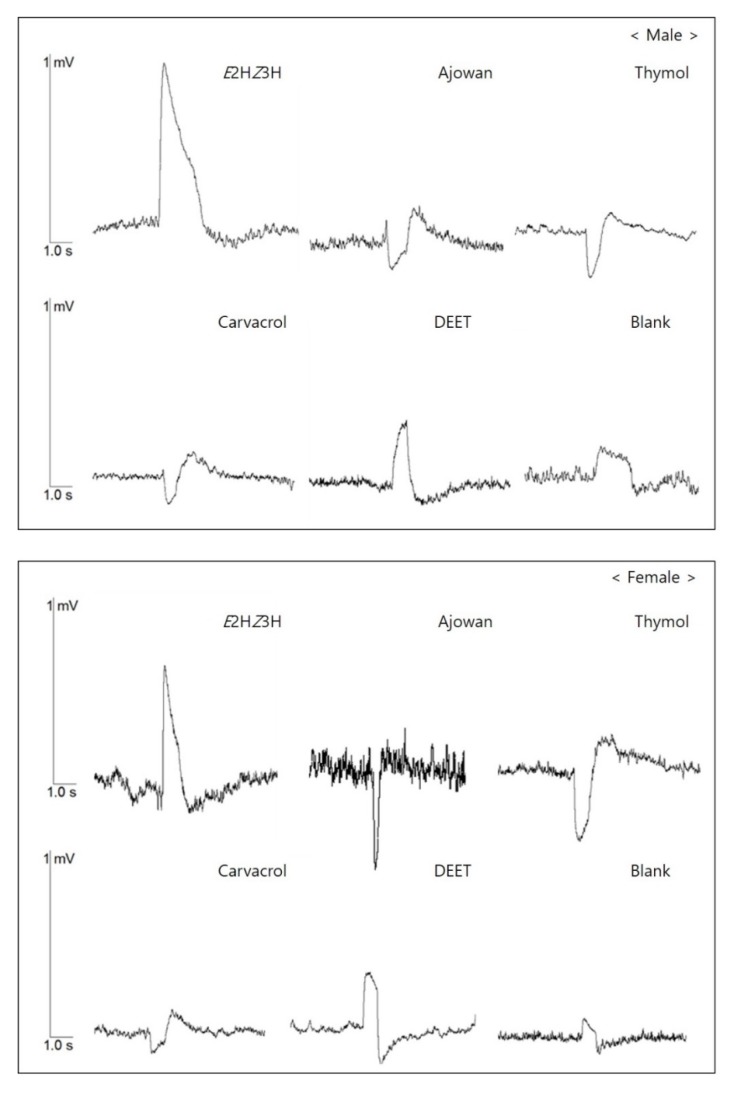
EAG response waveforms of adult bean bugs (**up**: male; **down**: female) elicited by *E*2H*Z*3H, ajowan essential oil, thymol, carvacrol and DEET.

**Table 1 insects-11-00104-t001:** Chemical composition of ajowan essential oil.

Compound	Retention Index		Composition Rate (%)
	DB-1MS	HP-INNOWAX	
*α*-Pinene	928	1021	0.87
Camphene	940	1064	0.10
*β*-Pinene	967	1108	1.26
Myrcene	981	1165	0.48
*α*-Terpinene	1007	1181	0.13
*p*-Cymene	1012	1275	24.40
1,8-Cineole	1018	1209	0.32
Limonene	1020	1200	0.44
*γ*-Terpinene	1050	1248	27.77
Terpine-4-ol	1159	1610	0.32
Thymol	1273	2207	41.77
Carvacrol	1278	2235	0.55
Sum	-	-	98.41

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
