# Peer review of "Behavioral and Electrophysiological Effects of Ajowan (Trachyspermum ammi Sprague) (Apiales: Apiaceae) Essential Oil and Its Constituents on Nymphal and Adult Bean Bugs, Riptortus clavatus (Thunberg) (Hemiptera: Alydidae)"

_insects, 2020, doi:10.3390/insects11020104_

Round 1
Reviewer 1 Report
Review
This is very tidy study that examines various aspects of the ajowan extract on bean bugs. The manuscript is well-written and contributes well to the literature. There are very minor suggestions for wording that sharpen the meaning of the english wording. Overall very well done and only minor changes are suggested.
Line 23 “oil constituents” not “oils constituent” or you could say ‘individual constituents of ajowan oil
Line 22, 28 and 127 etc. The phrasing of “until xxx ug” is awkward. You are trying to say “ up to the highest dose tested (14.15 ug/cm3)”. The use of “until” is more time-related. I would suggest a slight rephrasing for the instances where you used this phrase.
Line 128 “In adult male” would be better stated as “For adult males”.
Line 131 “until” could be ‘up to and including”
Line 133 ‘male’ should be “males” “repellent” should be “repellency” and “until” should be “up to”
Line 170 this is a bit tricky. I read the published paper and understand why you want the chromatogram in this manuscript. I think that you need to add a little bit more wording in to tie in the chromatogram with the identified compounds. For instance, you could say that your analysis agreed with a prior report (19) and that the 12 most abundant compounds identified from the ajowan extract and their abundance were presented in Figure 3. I would suggest a little bit more wording there that ties things together.
Line 198 concentrations
Line 210 Dermanyssus
Line 268 I would add the dose here (1mg) in the title as the dose is high and really should induce a large response which is does for these compounds. Also what is N? How many antennae were tested? You have means but I did not see how many were tested.
Line 255 The first sentence sort of sounds like there is a significant difference between males and females. I would suggest putting the phrase “between compounds” after difference.
Line 260 the species name for American cockroach should be lower case

Author Response
Reviewer 1
This is very tidy study that examines various aspects of the ajowan extract on bean bugs. The manuscript is well-written and contributes well to the literature. There are very minor suggestions for wording that sharpen the meaning of the english wording. Overall very well done and only minor changes are suggested.
Line 23 “oil constituents” not “oils constituent” or you could say ‘individual constituents of ajowan oil
: We changed according to reviewer’s comment.
Line 22, 28 and 127 etc. The phrasing of “until xxx ug” is awkward. You are trying to say “ up to the highest dose tested (14.15 ug/cm3)”. The use of “until” is more time-related. I would suggest a slight rephrasing for the instances where you used this phrase.
: We changed as “up to 14.15 μg/cm2. We changed this sentence through the manuscript.
Line 128 “In adult male” would be better stated as “For adult males”.
: We changed according to reviewer’s comment.
Line 131 “until” could be ‘up to and including”
We changed as “up to”
Line 133 ‘male’ should be “males” “repellent” should be “repellency” and “until” should be “up to”
: We changed according to reviewer’s comment.
Line 170 this is a bit tricky. I read the published paper and understand why you want the chromatogram in this manuscript. I think that you need to add a little bit more wording in to tie in the chromatogram with the identified compounds. For instance, you could say that your analysis agreed with a prior report (19) and that the 12 most abundant compounds identified from the ajowan extract and their abundance were presented in Figure 3. I would suggest a little bit more wording there that ties things together.
We described GC and GC-MS analysis in detail at Material and Methods section. We also changed Figure to Table. We also added sentence below as reviewer suggested.
“Our analysis of ajowan essential oil with GC and GC-MS agreed with a prior report [38], and that the 12 most abundant compounds identified from ajowan essential oil and their abundance were presented in Table 1.”
Line 198 concentrations
: We changed according to reviewer’s comment.
Line 210 Dermanyssus
: We changed according to reviewer’s comment.
Line 268 I would add the dose here (1mg) in the title as the dose is high and really should induce a large response which is does for these compounds. Also what is N? How many antennae were tested? You have means but I did not see how many were tested.
: We added dose at Figure caption. We added numbers of replication at Material and Methods section.
Line 255 The first sentence sort of sounds like there is a significant difference between males and females. I would suggest putting the phrase “between compounds” after difference.
: We changed according to reviewer’s comment.
Line 260 the species name for American cockroach should be lower case
: We changed according to reviewer’s comment.
Reviewer 2 Report
This manuscript explores the repellency of select essential oils and of the constituents that comprise ajowan essential oil to the bean bug Riptotus clavatus. They did this by utilizing a 4-arm olfactometer over a 24-hr period to record positional data of individual bean bugs. GC-MS was utilized to confirm the constituents of ajowan oil, and finally electroantennogram recordings were performed to validate olfactory reception and repellent response to tested compounds.
In general this manuscript needs to be improved. The authors need to address the limitations of this study further in respect to their conclusions. In addition, edits are required to improve the grammar and readability of the manuscript. Finally, improvements to the methods are needed.
See attached file for specific comments.

Author Response
Reviewer 2
Summary:
This manuscript explores the repellency of select essential oils and of the constituents that comprise ajowan essential oil to the bean bug Riptotus clavatus. They did this by utilizing a 4-arm olfactometer over a 24-hr period to record positional data of individual bean bugs. GC-MS was utilized to confirm the constituents of ajowan oil, and finally electroantennogram recordings were performed to validate olfactory reception and repellent response to tested compounds.
In general this manuscript needs to be improved. The authors need to address the limitations of this study further in respect to their conclusions. In addition, edits are required to improve the grammar and readability of the manuscript. Finally, improvements to the methods are needed. Below are specific comments to further address the above concerns.
Abstract
Lines 20-22: “For the…14.15μg/cm2 concentration.” This needs to be rewritten for clarity. Similar sentences are found throughout the manuscript. Using the word “until” leaves the impression that concentrations are increasing to 14.15μg/cm2, when the authors seemingly mean to indicate that significance is seen at concentrations above 14.15μg/cm2.
: We changed as “up to 14.15 μg/cm2. We changed this sentence through the manuscript.
Lines 22-23: change “We also investigated…ajowan essential oils constituents.” To “…ajowan essential oil constituents.”
: We changed according to reviewer’s comment.
Line 24: Change “…toward the…” to “…”against…”
: We changed according to reviewer’s comment.
Lines 27-28: Another instance of using “until” please change to improve the understanding of the findings. Additionally change “…the strong repellent…” to “…strong repellent…”
: We change “until” as “up to”. We changed this through the manuscript. We changed according to reviewer’s comment.
Lines 30-32: The statement here is repeated several times within the manuscript. While within the closed olfactometer assay there is an apparent repellent/deterrent to a treated area this study does not indicate that the tested compounds can be used as push components/stimul for a ‘push-pull’ system. It does indicate that these compounds are of interest for potential use as push stimuli however further studies are required to validate that these compounds would work outside of this closed system. The authors need to address this better within the discussion section of this manuscript.
Introduction
: We changed “can be used as push components/stimuli” as “can be potential candidates as push stimuli” throughout the whole manuscript.
Line 39: “…manage the bean bugs.” Should read “…manage bean bugs.”
: We changed according to reviewer’s comment.
Line 42: …”the spray of insecticides.” Should read “…insecticide application.”
: We changed according to reviewer’s comment.
Line 44: the word “monitor” needs to be corrected.
: We changed according to reviewer’s comment.
Lines 52-55: “Providing insects….control methods.” This sentence needs improvement. As it is, it reads as trap cropping are odors along with host plant volatiles and pheromones. Please revise to improve the clarity of the statement. : We deleted “trap cropping” in this sentence.
Materials and Methods
Lines 76-81: Is this a standard rearing method? And was it developed by the authors? If not is there a citation?
: We added the citation.
We reared the bean bugs according to Choi et al. [19] with slight modification.
Lines 83-85: Fairly limited description of the GC and GC-MS method. Even if following a previously described method more description is needed.
: We added description of GC and GC-MS method in detail at Material and Methods section.
Lines 87-107: Has this method been employed before? If so citation? Were there multiple discs placed into the assay at once? It states that the experimental paper disc were placed into one set of two opposite chambers. Does this mean that two different discs were placed into different chambers across from each other and then two control discs were placed into the remaining chambers?
For the lines 101-102 discussing reducing the effect of direction and position mean that control discs and test discs were switched.
Was light bias controlled for?
: We added citation for this method as below.
We evaluated the plant essential oils’ and constituents’ repellent activities for nymph and adult bean bugs using a four-arm olfactometer with a slight modification of prior study [20].
We placed treated two paper discs into one set of two opposite chambers and non-treated two paper discs into another set of two opposite chambers. Line 101-102: We used 10-30 four-arms olfactometers at the same time for each bioassay. If treated paper disc in half numbers of olfactometers placed in the same direction, treated paper disc in another half numbers of olfactometer placed in the opposite direction. Because several fluorescent lights placed at ceiling of insect rearing room in a regular grid of lamps, olfactometrs received uniform lighting. So light bias is minimum.
Line 100: “Individual…was used only once.” Should read “Individual…were used only once.”
: We changed according to reviewer’s comment.
Lines 112-119: This is another section that is lacking in detail for method. Additionally, there is not a citation here for the method.
: We described more detail and added citation.
We used an EAG set composed of a signal acquisition meter (IDAC-4, Syntech, Germany), probes (universal single ended probe, Syntech, Germany), and a stimulus controller (CS 55, Syntech, Germany) to record the electrophysiological response of the male and female adult bean bugs to ajowan essential oil, thymol, carvacrol, deet and (E)-2-hexenyl (Z)-3-hexenoate (E2HZ3H, the bean bug’s aggregation pheromone component). Before connecting the EAG probe to the bugs, we cut the basal and tip segments of males (n = 5) and females (n = 5) antennas. The universal probe attached glass capillary (1.1 mm I.D., paul marienfeld gmbh, Germany) filled with 0.1 N KCl was connected to the basal part of antenna and used as recording electrode. Another probe attached glass capillary filled with 0.1 N KCl was connected to the tip of antenna and used as reference electrode [21]. Test compounds (1 mg) dissolved with 10 uL of hexane were loaded onto a paper disc (8 mm, Advantec MFS, Inc., California, USA). The paper disc for control received hexane only. After evaporation of hexane for 30 sec, each test compound stimulated the antenna for 1.5 sec with filtered air using stimulus controller. Electrophysiological responses of antenna to each test compound was acquisited using signal acquisition meter and measured the magnitude of each responses using auto spike (Syntech, Germany). The data was analyzed using ANOVA and compared treatment mean values using Duncan’s test (IBM SPSS Statistics V25.0, Armonk, NY, USA).
Results and Discussion
Lines 124-125: “…nymphs at 226.35 μg/cm2 concentration.” Should read “…nymphs at a concentration of 226.35 μg/cm2.”
As a note there are several instances throughout the remainder of the manuscript where this edit can be made. Going forward when getting to those line it will read “see above” to indicate a similar edit is needed.
: We changed according to reviewer’s comment throughout a whole manuscript.
Lines 126-128: “The bean bug nymphs…” should read “Bean bug nymphs…” Additionally, this is another instance of using the word until.
: We changed according to reviewer’s comment throughout a whole manuscript.
Line 128: “In adult male,…” should read “In adult males,…”
: We changed “For adult males” as reviewer 1’s comment.
Line 168: what is meant by “…bean bug’s powerful egg parisitoids…”? Are they physically robust or are you meaning they are effective parisitoids? This is not clear.
: We changed as “effective parasitoids”
Lines 170-172: Not a complete paragraph as is. In addition a short summary of findings should be included from that study.
: We added sentences as below.
Our analysis of ajowan essential oil with GC and GC-MS agreed with a prior report [38], and that the 12 most abundant compounds identified from ajowan essential oil and their abundance were presented in Table 1. The most abundant compound was thymol (41.77%) followed by γ-terpinene (27.77%), p-cymene (24.40%), and β-pinene (1.26%). Composition rate of other compounds was less than 1%.
Lines 174-176: it needs to be clarified somewhere in the manuscript as to what “strong repellency” is. Repellent values are considered strong above X%. Without this saying a repellent is strong can be subjective without giving the reader a baseline of what the authors consider to be strong.
Further: “…thymol showed strong repellent for the…” should read “…thymol showed strong repellency against…”
Additionally, the remainder of the sentence talks only about carvacrol and the beginning of the sentence references both carvacrol and thymol. This needs to be rewritten for clarity.
: We changed sentence as below.
Significant difference in repellency was observed between the carvacrol-treated chamber and the untreated chamber up to 0.71 μg/cm2. The repellent activity of thymol was 70.0% at 2.83 μg/cm2 , but reduced to 63.3% at 1.41 μg/cm2. No significant difference in the repellency was observed between the chamber treated with the other ajowan essential oil constituents and untreated chamber at 22.64 μg/cm2.
Lines 177-178: “The thymol repellent rate…” should read “The repellent rate of thymol…”
“at 2.83 μg/cm2 concentration..” should read “at a concentration of 2.83 μg/cm2…”
Remove the last “concentration” word at the end of the sentence.
: We changed according to reviewer’s comment.
Lines 178-179: This sentence needs to be reworded. It currently reads as if the treated and untreated chambers are the things being repelled.
We changed as below.
: No significant difference in the repellency was observed between the chamber treated with the other ajowan essential oil constituents and untreated chamber at 22.64 μg/cm2.
Line 180: “...the bean bug…” should read “…bean bug…”
: We changed according to reviewer’s comment.
Lines 182-184: “The male bean bug…” should read “Male bean bug adults exhibited repellency, to the ajowan oil constituents, that was similar to what was seen in the nymphs.”
: We changed according to reviewer’s comment.
Lines 184-185: “Carvacrol repellent rates…” should read “The carvacrol repellent rate was 82.61 at 5.66 μg/cm2, and the rate reduced further to 59.26 at 2.83 μg/cm2.”
: We changed according to reviewer’s comment.
Line 186: Another instance of “until”, and see above.
: We changed “until” as “up to”.
Lines 187-188: See above
: We changed according to reviewer’s comment.
Line 194: “The female bean bug…”should read “Female bean bug…”
: We changed according to reviewer’s comment.
Lines 195-196: See above
: We changed according to reviewer’s comment.
Line 197: What constitutes low repellency? Similar to saying strong, this needs to have parameters for the reader.
We changed sentence as below.
Because repellent activity of α-pinene, 1,8-cineole, and terpinen-4-ol was weaker than those of thymol and carvacrol at 22.64 μg/cm2, we did not investigate their repellent activity at lower concentrations. Significant difference in repellency was observed between carvacrol- or thymol-treated chamber and untreated chamber up to 2.83 μg/cm2.
Lines 199-200: Another instance of using “until”
: We changed according to reviewer’s comment.
Line 200-201: Are the authors saying that females were not repelled by DEET? Or they were only significantly repelled by concentrations above 22.64 μg/cm2?
:We changed as “Significant difference in repellency was not observed between DEET treated chamber and untreated chamber at 22.64 μg/cm2.”
Concentration of 22.64 μg/cm2 is the highest dose for test compound, and female adult were not repelled by DEET at this dose.
Lines 216-217: “…against insect belonging to Hemiptera…” should read “…against other hemipteran insects has...”
: We changed according to reviewer’s comment.
Line 218: “…bean bugs were…” should read “…bean bugs was…”
: We changed according to reviewer’s comment.
Lines 218-219: The prior studies were all laboratory studies that provide evidence that thymol and carvacrol can repel several different insects, however none of them validate a push-pull system using these compounds. This needs to be addressed. :
We changed as below: This and prior studies indicate that thymol and carvacrol can be potential candidates as ‘push’ stimuli in push-pull strategies for bean bug control.
Lines 225-226: toxicity may not affect the parasitoid, but could these compounds repel the parasitoid? Can this have an affect on the effectiveness of its control of bean bug?
We added sentence as below.
However, the effect of ajowan essential oil, thymol and carvacrol on the behavior of egg parasitoids should be investigated in future study..
Lines 228-229: Citation is needed for the volatility statement. Also is it just volatility that makes it so that plant essential oils don’t work similarly in the field compared to laboratory assays? What about the weather, sunlight, etc..?
: We added the citation, and we changed sentence as below.
Several studies have reported many plant-derived materials exhibiting repellent properties in recent years [51]. However, nearly all plant-based repellents derived from plant essential oils have limited residual activity (<2–4 hours), primarily due to their high volatility [52,53].
We think key point for practical use of plant essential oil and their constituents is to develop suitable formulation for constant release.
Lines 228-234: The beginning statement indicates that plant essential oils don’t work well in the field due to volatility. The second part references a lab study that indicates thymol and carvacrol have longer residuals, however can this directly suggest that the compounds will be effective in the field?
: As described above, high volatility is one obstacle for practical use of plant essential oils and their constituents. Thymol and carvacrol have longer residual, and this could be advantage for practical use in filed compared with other constituents with high volatility.
Line 247: Consider relisting pheromones tested. Up until this point they were only referenced in the methods section and a reader may not remember what they are.
We changed the sentence as below.
Mean EAG responses of the male R. clavatus to E2HZ3H (the bean bug’s aggregation pheromone component), ajowan essential oil, thymol, carvacrol, DEET and the blank were 0.90 mV, 0.36 mV, 0.29 mV, 0.20 mV, 0.42 mV and 0.25 mV, respectively.
Lines 250-252: Were the amplitudes significantly higher or only the shapes of the EAG spikes? Also using the word much does not give any context to a reader. It is subjective like using very.
We added new Figure 4 that shows amplitude of each compound. The mean electrophysiological response were analyzed using ANOVA and compared treatment mean values using Duncan’s test.
Lines 273-275: Again stating that ajowan, thymol, and carvacrol can be used as a push stimulus in a push pull system is not a valid statement based on the results here. A push pull system was not tested, however these compounds could prove useful in such a system.
: We changed sentence as below.
Based on our results, we could conclude that ajowan plant essential oil and its constituents carvacrol and thymol can be potential candidates as the ‘push’ component with an aggregation pheromone trap as the “pull” component in a “push-pull” strategy of bean bug control.
Reviewer 3 Report
Through the whole manuscript, change the term 'repellent rate' to 'repellency' or 'repellent activity', since 'rate' is defined the change in given time period. The number of significant figure should be shortened, for example, 82.61% -> 82.6%. Line 69, reagents -> test compounds, since there is no reaction involved. Figure 2, scientific name should be present in Italic, and add comma at Corinader, seed. Figure 3 should be presented as a table, and no need for the chromatogram chart. The authors should use more clear (higher resolution) charts for figure 5.
Author Response
Through the whole manuscript, change the term 'repellent rate' to 'repellency' or 'repellent activity', since 'rate' is defined the change in given time period. The number of significant figure should be shortened, for example, 82.61% -> 82.6%. Line 69, reagents -> test compounds, since there is no reaction involved. Figure 2, scientific name should be present in Italic, and add comma at Corinader, seed. Figure 3 should be presented as a table, and no need for the chromatogram chart. The authors should use more clear (higher resolution) charts for figure 5.
: We changed repellent rate to repellent activity through the whole manuscript. We shorten the significant figure through the whole manuscript. We change reagents to test compounds. We change scientific name in Italic. We added comma at Coriander, seed. We changed Figure 3 to Table 1. We improved Figure quality.
Round 2
Reviewer 2 Report
The manuscript has been improved to allow for publication, however it still requires a minor edit.
Lines 21-22 in the abstract.
The authors changed the wording from "until 14.15 µg/cm2" to "up to 14.15 µg/cm2".
This essentially means the same thing and is incorrect in describing what is in Table 2.
The way this reads using the phrase that the authors do suggests that there are concentrations below 14.15 µg/cm2 that were utilized. According to Table 2, 14.15 µg/cm2 was the lowest concetration used for Ajowan oil testing. So what the authors would want to say is "significance is seen at concentrations above 14.15 µg/cm2" or they can say "down to 14.15 µg/cm2". The use of "up to" will need to be corrected throughout the manuscript.
Additionally, lines 21-22 in the abstract state that Ajowan oil was significantly repellent to adult bugs down to 14.15 µg/cm2, however looking at Table 2 female adult bed bugs were still significantly repelled at 14.15 µg/cm2. This statement needs to be corrected to exclude females as nymphs and males were no longer repelled signifcantly at this concentration.
Author Response
Dear Editor of Insects,
Thank you for considering this manuscript (insects-693802) so promptly and carefully, and for very valuable comments; I believe that I have been able to address all of them. The answers to your comments are indicated below, and the manuscript has been adjusted accordingly. All changes are marked in red. I hope that you will agree that these changes adequately address the queries that have been raised, and hope that you will find it to be acceptable now for publication.
Yours Sincerely,
Prof. Il-Kwon Park
Reviewer 2
The manuscript has been improved to allow for publication, however it still requires a minor edit.
Lines 21-22 in the abstract.
The authors changed the wording from "until 14.15 µg/cm2" to "up to 14.15 µg/cm2".
This essentially means the same thing and is incorrect in describing what is in Table 2. The way this reads using the phrase that the authors do suggests that there are concentrations below 14.15 µg/cm2 that were utilized. According to Table 2, 14.15 µg/cm2 was the lowest concetration used for Ajowan oil testing. So what the authors would want to say is "significance is seen at concentrations above 14.15 µg/cm2" or they can say "down to 14.15 µg/cm2". The use of "up to" will need to be corrected throughout the manuscript.
: We changed “up to” as “down to” throughout the manuscript.
Additionally, lines 21-22 in the abstract state that Ajowan oil was significantly repellent to adult bugs down to 14.15 µg/cm2, however looking at Table 2 female adult bed bugs were still significantly repelled at 14.15 µg/cm2. This statement needs to be corrected to exclude females as nymphs and males were no longer repelled signifcantly at this concentration
: We changed the sentence as below.
“For female adults, the repellent levels were significantly different between ajowan oil from a treated chamber and from an untreated chamber down to 14.15 μg/cm2.”